# Comparison Research on Characterization and Evaluation Approaches for Paint Coated Corrosion Using Eddy Current Pulsed Thermography

**DOI:** 10.3390/s23156889

**Published:** 2023-08-03

**Authors:** Chao Xue, Yinqiang Zhang, Song Ding, Cheng Song, Yiqing Wang

**Affiliations:** 1School of Electrical Engineering and Control Science, Nanjing Tech University, Nanjing 211816, Chinawangyiqing@njtech.edu.cn (Y.W.); 2Key Laboratory of Nondestructive Testing, Fujian Polytechnic Normal University, Fujian Province University, Fuqing 350300, China; 3Siemens Limited China, Beijing 100102, China; 13911899416@139.com

**Keywords:** corrosion evaluation methods, paint coated corrosion, eddy current pulsed thermography, smart sensor system

## Abstract

Paint coated corrosion detection and evaluation is a big challenge for steel performance and structure health. Eddy current pulsed thermography (ECPT) technique is investigated because it can reflect the corrosion physical properties through paint coating by the infrared signal. This paper proposes skewness method, which presents the feature of temperature curve’s shape automatically, and compares it with principal component analysis (PCA), phase analysis, and kurtosis feature extraction methods for paint coated corrosion characterization and evaluation. The averaged skewness shows the best sensitivity for 0–6 months corrosion. The normalized second principal component (PC) presents good sensitivity and the best measurement scale for corroded time. Furthermore, the temperature curve analysis proves that the electrical conductivity dominates the induced heating and heat distribution. The corrosion height is utilized to explain why ECPT technique is valid within 10 months corroded time. ECPT technique is proved as a smart sensor system for paint coated corrosion detection and characterization.

## 1. Introduction

Steel is a fundamental material for our life and industry. However, corrosion, the wastage and destruction caused by chemical or electrochemical reactions with surroundings, weakens steel’s performance and damages the structure’s health, even human lives [1,2]. For corrosion measurement, many physical, chemical, and electrochemical methods were employed, such as mass loss measurement, scanning Kelvin probe (SKP), ultrasonic testing (UT), scanning acoustic microscopy (SAM), electrochemical impedance spectroscopy (EIS), electrochemical noise (EN), and so on [3,4,5,6]. With the wide application of the paint coating technique, which is provided to protect steel or structure surfaces, it is more difficult to detect the local corrosion under the paint coating. In order to non-destructively detect local corrosion under paint coatings, there are some non-destructive testing techniques, such as the Barkhausen noise method, eddy current testing method, magnetic field measurement method, etc. However, these technologies are susceptible to factors such as material magnetoelasticity, texture, and stress distribution, making signal analysis more complex, and eddy current testing can only detect defects on the surface or near the surface of the material. Eddy current pulse thermal imaging technology can utilize multidimensional feature fusion algorithms to improve defect recognition rate and quantitative evaluation ability [7].

Recently, the ECPT technique is engrossing as well as the corresponding signal processing methods [8,9,10]. For this technique, an AC supply is employed to drive an excitation magnetic field, which generates induced eddy current and Joule heat on the surface of the conductive material. After a short time, the power supply is switched off and the specimen’s temperature falls to the initial value. The two stages, including Joule heat generation and temperature decline are recorded by an infrared camera. As the infrared emission could reflect the physical properties and their variation of the specimen, the ECPT technique was investigated for nondestructive testing and characterization for defects, such as cracks and corrosion blister. This novel technique has the advantages of non-contact and mobile, big lift-off distance, high sensitivity and resolution, which contributes to on-site detection and characterization. In particular, the excited magnetic field can induce an eddy current on the steel through a nonconductive paint layer, which makes it possible to detect defects under paint coating.

For processing the infrared images, transient analysis methods and statistic algorithms are investigated. M. Pan et al. [11] employed PCA and independent component analysis (ICA) [12] methods to characterize the delamination defect of carbon fiber reinforced plastic (CFRP). The PC and independent components were extracted as the features for thermal image reconstruction and damage characterization. Y. He et al. [13] compared PCA, ICA and fast Fourier transform (FFT) in the performance of blister detection. In conclusion, the third PC, and the first and fourth independent components could be used to detect corrosion blisters. The FFT algorithm could eliminate the non-uniform heating and detect the blister easily. R. Yang et al. [14] analyzed the process of joule heat generation, conduction and radiation on the corroded and uncorroded regions. Accordingly, a heat transport model and phase analysis approach were proposed. The phase of temperature variation for each pixel was calculated as the feature by the FFT method. The proposed approach could characterize the corrosion region and nondestructively evaluate different corrosion states from 1 month to 6 months.

Compared with the transient methods, phase analysis extracts the features in the frequency domain based on a single pixel, which eliminates the influence of non-uniform heating and emissivity difference of different locations on the sample surface. However, the phase deference is theoretically limited between −π~+π, and the frequency selection for analysis is manual. In order to provide an approach for practice, which not only characterizes the corrosion with the reconstructed image but also quantitatively measures the corrosion stage. This paper aims to investigate and compare the feature extraction methods and proposes an approach based on the extracted features of pixels in the marked region for quantitative corrosion stage characterization and evaluation.

In the rest of this paper, the principle of the ECPT system and different four feature extraction algorithms are introduced in Section 2. The sample preparation and experimental setup are illustrated in Section 3. Next, in Section 4, the feature extraction and reconstruction are presented, and the corrosion characterization performance of the proposed four algorithms is compared. Based on the analysis and discussion, the conclusion and feature work is proposed in the last section.

## 2. Methodologies

ECPT is a multi-physics coupled nondestructive testing and evaluation technique, including induction heating, thermal conduction, infrared image acquisition, and data processing [8,9,10]. For the ECPT system, as shown in Figure 1, an alternating current drives the excitation coil to induce a high-frequency eddy current in the conductive samples. Then, Joule heat is generated in a short time by the inductive eddy current and resistance of the material. Then, the infrared image sequence is acquired by an infrared camera to reflect the heat generation, conduction, and diffusion processes [15]. On the other hand, the physical properties of materials, such as electrical conductivity, thermal conductivity, permeability and emissivity, are closely related to the microstructure of the material. Corrosion gradually changes the composition and microstructure of the material, which causes complex changes in electrical conductivity, and thermal conductivity [16,17], which is governed by Equations (1) and (2).
(1)$$Q_{1}=\frac{1}{\sigma_{1}}{\left|J_{s}\right|}^{2},\;Q_{2}=\frac{1}{\sigma_{2}}{\left|J_{s}\right|}^{2}$$
(2)$$\rho_{1}C_{p1}\frac{\partial T}{\partial t}-\nabla \left(k_{1}\nabla T\right)=Q_{1},\;\rho_{2}C_{p2}\frac{\partial T}{\partial t}-\nabla \left(k_{2}\nabla T\right)=Q_{2}$$
where $Q_{1}$ is the induced Joule heat of the corrosion layer, $Q_{2}$ is the induced Joule heat of the uncorroded layer, $\sigma_{1}$ is the electrical conductivity of the corrosive layer, $\sigma_{2}$ is the electrical conductivity of the uncorroded layer, $J_{s}$ means eddy current density. $\rho_{1}$ is the material density of the corrosive layer, $\rho_{2}$ is the material density of the uncorroded layer, $C_{p1}$ is the material heat capacity of the corrosive layer, $C_{p2}$ is the material heat capacity of the uncorroded layer, $k_{1}$ is the thermal conductivity of the corrosive layer, $k_{2}$ is the thermal conductivity of the uncorroded layer. The degree of corrosion, as well as the properties of the products, can cause changes in conductivity and thermal conductivity. Therefore, the temperature variation indicates the state of the surface.

Considering the skin effect, described as Formula (3), the higher excitation frequency means the shallower depth of eddy current distribution, therefore, as well as the Joule heat distribution. Thus, the initial thermal layer can be estimated by δ [18]
(3)$$\delta =\frac{1}{\sqrt{f\sigma \pi \mu }}$$
where f is the frequency of excitation, σ and μ are the electro-conductivity and permeability of the conductive material, respectively. Considering the excitation frequency and induction time, the initial heat flux on the sample surface is limited. Detailed information on these two parameters will be displayed in the experimental device.

Furthermore, due to the thermal image sequence including the noise of environmental temperature, sensor array, and sample geometry, feature extraction algorithms should be utilized to eliminate the test noise and enhance the accuracy.

### 2.1. Fast Fourier Transform

According to the model proposed by R. Yang [14], thermal waves are generated during heat conduction between corroded and free areas, and any waveform can be approximated by the sum of pure harmonics of oscillation at different frequencies. Due to the difference in the phase of the thermal pulse waveform between the corroded area and the uncorroded area, the Fourier transform algorithm can be used to extract the feature.

Fourier transform is a mathematical tool between the time domain and frequency domain. It is a commonly used method of analyzing signals, which essentially aggregates information from the time domain into the frequency domain. Any waveform, whether periodic or aperiodic, can be approximated by the sum of pure harmonics oscillating at different frequencies. In pulse eddy current thermal imaging, a pulse signal is applied to the coil, which excites the surface of the tested object to generate eddy currents. For an ideal pulse signal with zero duration, it can be decomposed into all frequency components of 0–∞, while the frequency components of the actual applied pulse signal are limited. The Joule heat generated by eddy current will also conduct thermal diffusion with these limited frequency components. The temperature distribution formed by diffusion is called a heat wave. According to the heat wave theory, the high-frequency heat wave propagates closer and faster, while the low-frequency heat wave propagates farther and slower. Therefore, low-frequency excitation can reveal deeper structural information within the tested object. However, the amplitude and phase of thermal waves are not only affected by pulse excitation within the tested object but also by changes in the thermal diffusion coefficient caused by defects. Therefore, defect information can be extracted from the amplitude-frequency and phase-frequency plots [19].

The infrared radiation signal collected by an infrared camera is a discrete signal. Therefore, a discrete Fourier transform is needed to decompose the signal into different frequency components. In this paper, the temperature response of each pixel is calculated according to the following one-dimensional discrete Fourier transform formula [20]:(4)$$F_{n}=∆t\sum_{k=0}^{N-1}T\left(k∆t\right)e^{-\frac{i2\pi nk}{N}}={Re}_{n}+{Im}_{n}$$
where ∆t is the sampling interval, n designates the frequency increment, N is the number of samples, ${Re}_{n}$ and ${Im}_{n}$ are, respectively, the real and imaginary components of $F_{n}$. Then, the amplitude and phase can be calculated by the following equation [20]: (5)$$A_{n}=\sqrt{{Re}_{n}^{2}+{Im}_{n}^{2}}$$
(6)$$\varphi_{n}={tan}^{-1}\left|\frac{{Im}_{n}}{{Re}_{n}}\right|$$

Then, the amplitude and phase of all pixels at certain frequencies are extracted to reconstruct the thermal images.

### 2.2. Principal Component Analysis

Considering the data acquisition of the ECPT system, the infrared camera transforms the infrared emission into a thermal image sequence and transmits it to a computer for further analysis. For a single thermal image, the value for each pixel signifies the temperature of an independent region. Because any object that is above absolute zero will release infrared radiation, the measured temperature of each pixel should be considered as a random signal that includes the noise of the sensor unit and environmental temperature variation. Therefore, the thermal image sequence can be seen as a 3D random data set. 

The PCA method was commonly used for reducing data dimension and noise [10]. Since the fundamental process of PCA is to calculate the orthorhombic directions in which the random data have obvious variances [20], as shown in Figure 2, this method is considered to have the potential to indicate the temperature gradient caused by defects. 

Due to the corrosion changes in the electrical conductivity, thermal conductivity, and permeability, the temperature distribution around the defect is uneven. PCA extracts the image region information with large temperature gradients, which cause obvious deviation and contain the features of defects in the PCs. Therefore, the position of corrosion can be characterized by the feature image reconstructed by the PCs.

### 2.3. Skewness and Kurtosis

For the whole process of thermal generation and heat dissipation, within the region where the induced eddy current exists, the surface temperature rises and decreases. The thermal image sequence can be regarded as the sampling for the temperature of the specimen’s surface. Therefore, for a certain pixel P, the pixel value in every thermal image makes up a temperature variation curve, as shown in Figure 3.

Considering the temperature variation of each pixel is closely related to whether it is corroded or not, the curve should be investigated for corrosion characterization. Therefore, this paper extracts features from the shape of the temperature curves to indicate the surface state. The shape characteristic of the temperature curve can be expressed by two statistical features: skewness and kurtosis.

As shown in Figure 4, the skewness is greater than zero when the data are located to the right of the mean, i.e., negative skewness; less than zero when the data are concentrated on the left side, called positive skewness. When the data are in the normal distribution, the skewness is 0 [22]. The skewness is the ratio of third-order center distance to variance. The formula is as follows [23].
(7)$$Skewness=\frac{n\sum {\left(x_{i}-\bar{x}\right)}^{3}}{(n-1)(n-2){sd}^{3}}$$
where n is the number of the data, $x_{i}$ is the data value, $\bar{x}$ means the average value of data, sd represents the standard deviation.

The temperature variation is different between corroded and uncorroded regions. Therefore, the skewness value of the temperature curve at each point of the sample surface is calculated, and an image is reconstructed to indicate the region of corrosion. Similarly, kurtosis is used to describe the distribution of data. It is investigated in this paper for characterizing the paint-coated corrosion by extracting the feature of temperature variation.

As shown in Figure 5, positive kurtosis indicates that the data are distributed near the average value, and negative kurtosis shows that the data are more dispersed and the curve shape is shorter and fatter. In mathematics, kurtosis is the ratio of the fourth-order central moment to standard deviation, and the calculation formula is as follows [25]:(8)$$kurtosis=\frac{\sum {\left(x_{i}-\bar{x}\right)}^{4}}{(n-1){sd}^{4}}-3$$

In this paper, PCA, phase analysis, skewness and kurtosis are employed for thermal image sequence feature extraction. The extracted feature values are reconstructed into an image, which facilitates the recognition of corrosion, uncorroded, heating and non-heating regions, for subsequent corrosion evaluation.

## 3. Specimens and Experimental Setup

Low carbon steel S275, which is widely used in the engineering field, is selected for corrosion samples. The composition of S275 is listed in Table 1. Samples are prepared in the size of 300 × 150 × 3 (length × width × height) mm^3^. These samples are covered by anticorrosive paint except the central area (30 × 30 mm^2^) to keep the steel beneath clean and dry. Then, the samples are exposed to a marine atmosphere at Blyth UK for a certain time (1, 3, 6, 10 and 12 months). After a certain exposure time, local corrosion forms on the area without anticorrosive paint covered. Images of coated and uncoated rust patches are shown in Figure 6.

The experimental apparatus is shown in Figure 7. EasyHeat224 from Cheltenham Induction Heating provides 380A_rms_ AC excitation current. According to previous studies, the electrical conductivity and relative permeability of steel are 4.68 × $10^{6}$ S/m and 60, while the corrosion products are close to 0.75 × $10^{6}$ S/m and 4 [12]. For eddy current generation, a rectangular coil is made of a 6.35 mm hollow copper tube, which connects to a water pump for cooling. The excitation frequency is set to 260 khz to generate heat on the sample surface. Based on this frequency, the initial heated region is controlled within 60 μm. The excitation time is limited to 200 ms to avoid obvious temperature rise.

A rectangular excitation coil is made of a 6.35 mm hollow copper tube, which is connected to a pump for water-cooling. An uncooled infrared camera with a 640 × 480 array of 17 μm detector, FLIR A655SC, from FLIR Systems, Inc., Wilsonville, OH, USA, is utilized to acquire IR image sequence with 50 hz sampling frequency and 30 mk temperature sensitivity. Considering the cooling process with no extra device, the whole record period is set to 2 s. A signal generator, Keysight 33500B, from Keysight Technologies Inc., Santa Rosa, CA, USA, is employed to generate a square pulse for synchronizing the excitation current and IR camera acquisition. The image sequence recorded by the infrared camera is transmitted to the PC, and then the four algorithms are performed and compared by MATLAB 2021b.

## 4. Results and Discussion

Firstly, the acquired infrared thermal imaging sequence is processed using the FFT algorithm with MATLAB. The reconstructed amplitude image and phase image are obtained, respectively. As shown in Figure 8, it is the amplitude image and phase image for 1, 3, 6, 10 and 12 months corroded samples using FFT processing. According to R. Yang’s research, the amplitude image at 5 hz has the best performance in the amplitude images. However, it is difficult to characterize the corrosion shape clearly. The better effect in phase image is 4 hz, which can easily determine the shape of corrosion (the red area of the middle square is the corrosion area). Therefore, this paper will only analyze and discuss the phase frequency image of 4 hz to compare with the reconstructed image of other methods.

For PCA methods, the sum of contribution rates of the first and second PCs is close to 90%, which means the first two PCs contain the most information of the original thermal image sequences. Therefore, only the first two PCs are analyzed and discussed. The reconstructed images of the first and second PCs for all the samples are shown in Figure 9. In this figure, (a), (c), (e), (g), (i) are the images of the first PC, which can hardly detect the corrosion area. However, (b), (d), (f), (h), and (j) are the reconstructed images of the second PC, which can clearly characterize the shape of the corrosion. Therefore, later in this paper, only the images of the second PC of PCA will be analyzed and discussed to compare with other methods.

The results of skewness analysis and kurtosis analysis based on temperature change curve analysis are shown in Figure 10. In this figure, (a), (c), (e), (g), and (i) are the corrosion images of 1, 3, 6, 10 and 12 months processed by the skewness method, respectively; (b), (d), (f), (h), and (j) are 1, 3, 6, 10 and 12 months corrosion images processed by kurtosis. The thermal imaging reconstitution processed by the skewness method can clearly characterize the corrosion shape, while the thermal imaging reconstitution processed by the kurtosis method cannot clearly characterize the corrosion. Based on the comparison of reconstructed images, this paper only discusses the images of the skewness method.

Among the proposed methods, FFT, PCA, and skewness can identify the corrosion area by the extracted features. While kurtosis only can indicate a vague shape of the corroded region for 6-month and 10-month samples. Furthermore, in order to compare the performance of the features for corrosion evaluation, the phase, second PC, and skewness are selected for further investigation.

Considering the infrared thermal imaging data are interfered with by the instrument itself, environmental temperature and other factors, the features in the corroded region (region ① in Figure 11) and uncorroded region (region ② in Figure 11) are averaged, respectively, based on the reconstructed images. The square of region ① has 30 × 80 pixels. Region ② is the uncorroded region, which is processed as well as region ①. The averaged features are shown in Table 2. All of the averaged features for corroded and uncorroded regions are plotted in Figure 12a. Furthermore, the averaged features are normalized between 0 to 1 for comparison as shown in Figure 12b.

From Figure 12a, the averaged second PC has the smallest variation from the uncorroded sample to the 10-month corroded sample, while the averaged skewness has the most obvious change for the same samples. That means the skewness feature has the best sensitivity for corrosion time evaluation, especially from 0 to 6 months of corroded time. As shown in Figure 12b, the normalized three features present non-monotonic variation versus corroded time. Averaged phase and second PC features decrease from an uncorroded state to 10 months corrosion, while increasing from 10 to 12 months corroded time. Averaged skewness decreases from 0 to 6 months of corroded time, but increases from 6 to 12 months of corroded time. Therefore, the selected three features can be used to evaluate the corrosion state before 6 months of corroding time. In particular, the averaged phase and second PC can recognize the corroded time till 10 months.

According to ref. [26], the induced heat power ($P_{h}$) is governed by:(9)$$P_{h}\propto I_{e}^{2}\sqrt{\frac{\mu f}{\sigma }}\;,\;\;\;\sigma =\frac{\sigma_{0}}{1+\alpha (T-T_{0})}$$
where $I_{e}$ is the excitation current, μ and f are the samples’ magnetic permeability and excitation frequency, respectively. σ is the electrical conductivity, which is related to temperature T and $T_{0}$. α and $\sigma_{0}$ are the temperature coefficient of resistivity and the electrical conductivity at $T_{0}$, respectively. Considering the temperature variations of this experiment below is at most 0.72 °C, the Formula (9) can be simplified as $P_{h}\propto k\sqrt{\frac{\mu }{\sigma_{0}}}$. With the experimental setup, $k=I_{e}^{2}\sqrt{f}$ is a constant. Based on Formula (2) and $W=P_{h}\cdot t$, which generates the Joule heat, Formula (9) can be rewritten as Equation (10) in the same heating conditions.
(10)$$T\propto k\sqrt{\frac{\mu }{\sigma_{0}}}$$

That means the magnetic permeability increases and electrical conductivity decreases leading to a temperature rise. For investigating the corrosion properties effect on the temperature variation, the average temperature of the uncorroded region (region ② in Figure 11) and corroded region (region ① in Figure 11) at 200 ms (the highest temperature) and 0 ms (non-heating temperature) are computed. Then, the maximum temperature variation can be calculated for comparison in Table 3.

From Table 3, for 1 month and 3 months, the maximum temperature variations of the uncorroded region are below the corroded region. Considering the oxide corrosion model, ferrous oxide (FeO) occurred first on the surface, which is soon covered by a stable ferric oxide (Fe_2_O_3_) layer. This oxide layer has very low electrical conductivity and permeability at room temperature [14,27]. After that, a ferromagnetic layer with the composition Fe_3_O_4_ formed between the previous oxide layers, which has obviously high electrical conductivity and permanent magnetism [28]. The initial corrosion products, goethite α−FeOOH and hematite $\alpha -{\mathrm{Fe}}_{2}O_{3}$, have lower electrical conductivity than steel. Also, hematite shows weak ferromagnetism, and goethite is anti-ferromagnetic at room temperature [23,29]. According to Equation (10), in the early stage, the corroded region has a more obvious temperature increase. However, over time not only does the magnetic permeability and electrical conductivity change but also the corrosion height alters, as shown in Figure 13, there are complex heat distributions for long-term corrosion. Comparing the data in Table 3, 6 months corroded sample has a higher temperature variation of uncorroded region than that of corroded region. As shown in Figure 13d, the height of 6 months corrosion is about 110 μm. Due to the electrical conductivity and permeability of corrosion products are significantly smaller than steel [23], which leads to a deeper heat distribution and lower surface temperature. Comparing the temperature variation of 3 months and 6 months of corrosion, we believe that electrical conductivity decrease dominates the Joule heat generation in early corrosion. 

However, for long-term corrosion, the situation is much more complex. Figure 13c presents a more complicated surface state than that of the early corrosion stage. Material is loose and flakes off making Joule heat generated, distributed, and conducted in a different layer, which means the infrared information can hardly characterize the corrosion’s physical properties. That is why the previous research only investigated the early corrosion before 6 months of exposure time.

## 5. Conclusions and Feature Work

In this paper, eddy current pulsed thermography is proven to be used to detect metal corrosion undercoating. Since the uneven heating and the emissivity anisotropy effect influence the thermal image acquisition, four feature extraction algorithms, FFT, PCA, skewness and kurtosis, are applied to compare the performance of feature extraction and corrosion evaluation. The main conclusion can be summarized as follows. 

(1)Although the first principal component has more contribution, the reconstructed image of the second PC can reflect the characteristics of corrosion better.(2)Despite the complex influence of corrosion properties and height, phase, skewness, and second PC can detect and recognize the early corrosion before 6 months. Especially, the averaged skewness has the most sensitivity among these corrosion evaluation approaches.(3)The normalized second PC has the best performance for corrosion evaluation. It presents almost the same sensitivity as normalized skewness and a bigger measurement scale, 0 to 10 months corrosion.(4)Based on the ECPT technique, all of these approaches can only evaluate the early-stage corrosion except PCA. Because the electrical conductivity dominates Joule heat generation and distribution (in depth) for early-stage corrosion. However, for long-term corrosion (more than 10 months exposure time), the height of corrosion means the infrared signals hardly characterize the physical properties of corrosion.

Considering the complex influence of physical properties variation on the sample surface infrared radiation, the proposed approach based on the ECPT technique can hardly characterize the long-time corrosion state. Some nondestructive testing methods that are based on a single physical principle, such as magnetic Barkhausen noise (MBN), and the eddy current test (ECT), can reflect the simple relations between corrosion state and magnetic or electrical properties. Therefore, with the advantages of data fusion and multi-parameter characterization, the ECPT technique combined with the MBN or ECT method can be the feature work for improving the corrosion state characterization and evaluation.

## Figures and Tables

**Figure 1 sensors-23-06889-f001:**
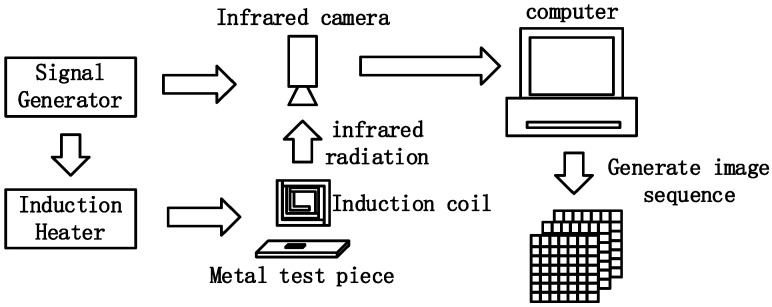
The diagram of ECPT system.

**Figure 2 sensors-23-06889-f002:**
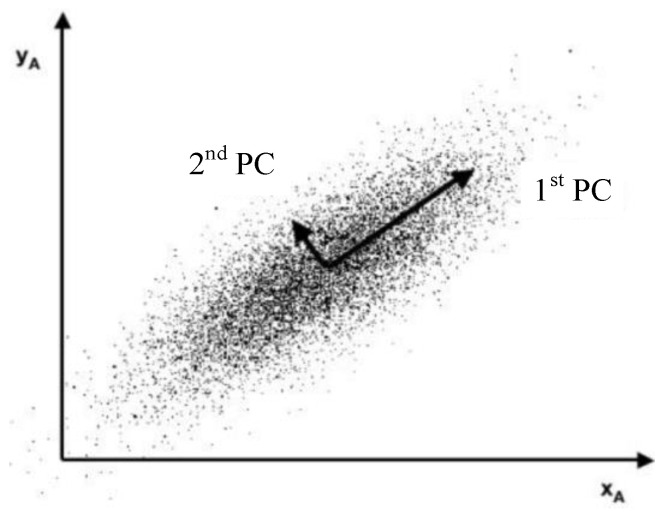
Schematic of feature extraction using PCA method [21].

**Figure 3 sensors-23-06889-f003:**
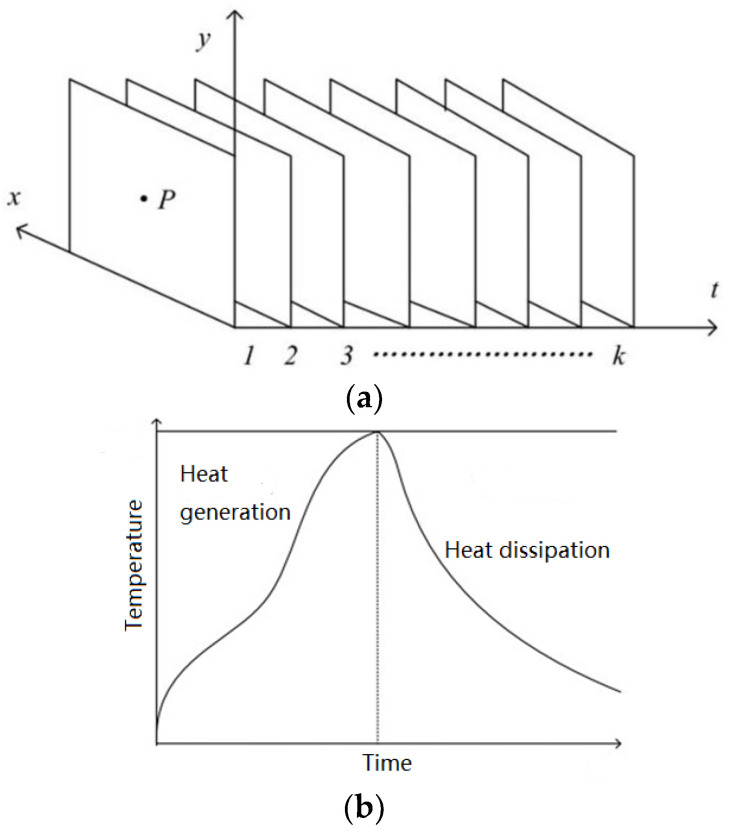
(**a**) Schematic of thermal image sequence, (**b**) temperature variation curve of pixel P.

**Figure 4 sensors-23-06889-f004:**
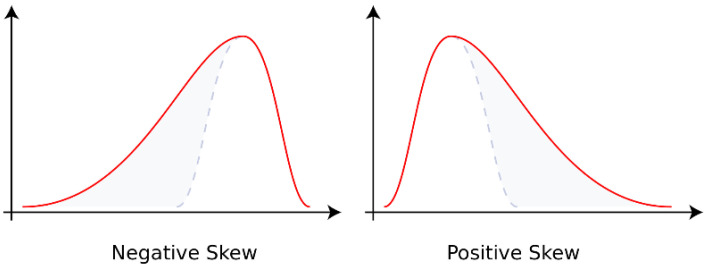
Schematic diagram of skewness. The blue dotted line means normal skewness [24].

**Figure 5 sensors-23-06889-f005:**
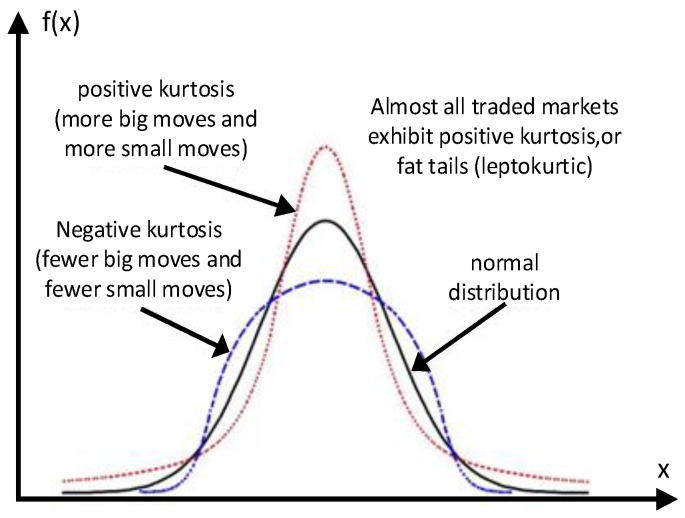
Schematic diagram of kurtosis.

**Figure 6 sensors-23-06889-f006:**
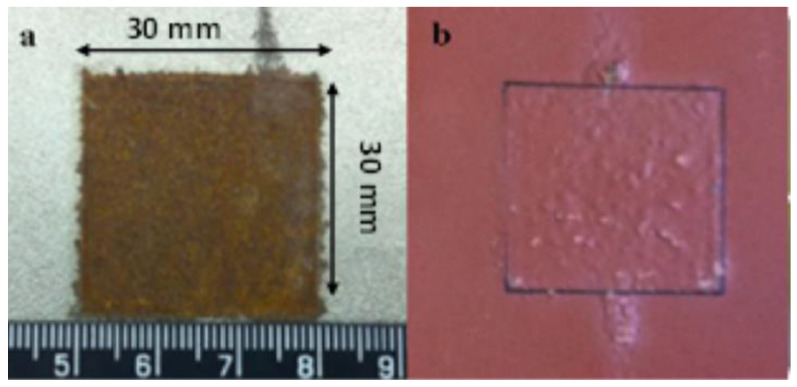
The images of (**a**) 1 month uncoated sample, (**b**) 1 month coated rust patch.

**Figure 7 sensors-23-06889-f007:**
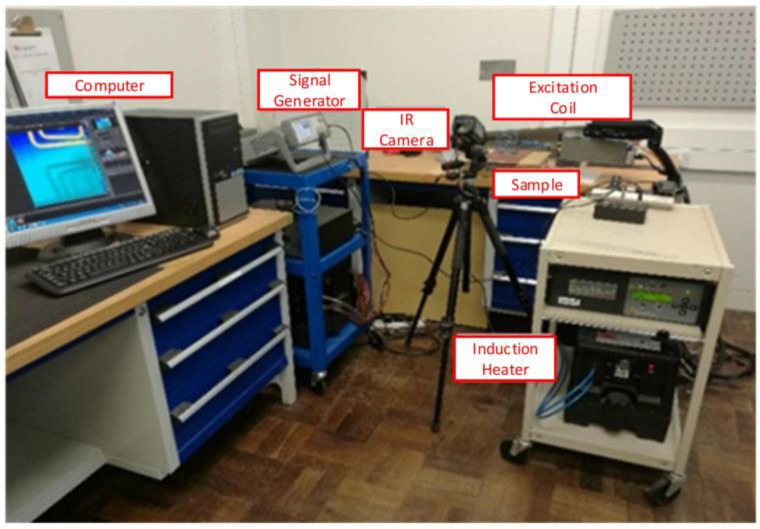
The experiment system.

**Figure 8 sensors-23-06889-f008:**
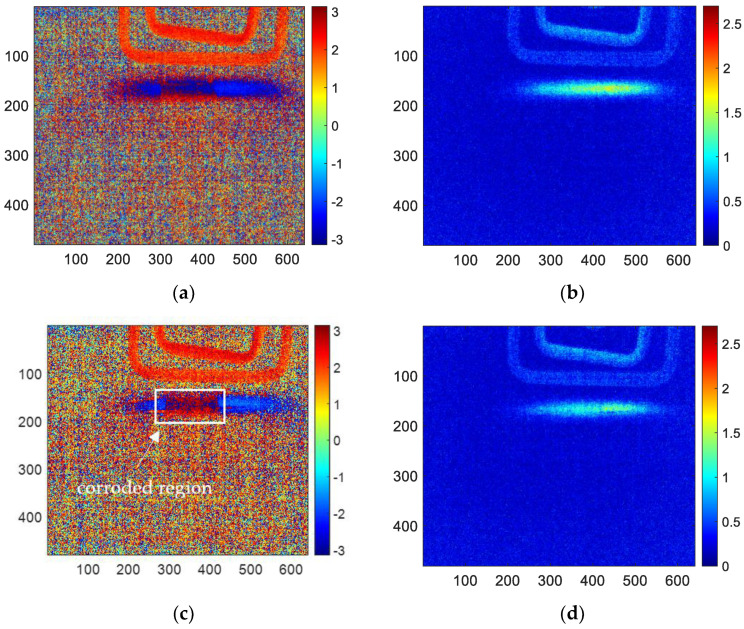
(**a**) 4 hz phase-frequency image of 1-month corrosion, (**b**) 5 hz amplitude-frequency image of 1-month corrosion, (**c**) 4 hz phase-frequency image of 3-month corrosion, (**d**) 5 hz amplitude-frequency image of 3-month corrosion, (**e**) 4 hz phase-frequency image of 6-month corrosion, (**f**) 5 hz amplitude-frequency image of 6-month corrosion, (**g**) 4 hz phase-frequency image of 10-month corrosion, (**h**) 5 hz amplitude-frequency image of 10-month corrosion, (**i**) 4 hz phase-frequency image of 12-month corrosion, (**j**) 5 hz amplitude-frequency image of 12-month corrosion.

**Figure 9 sensors-23-06889-f009:**
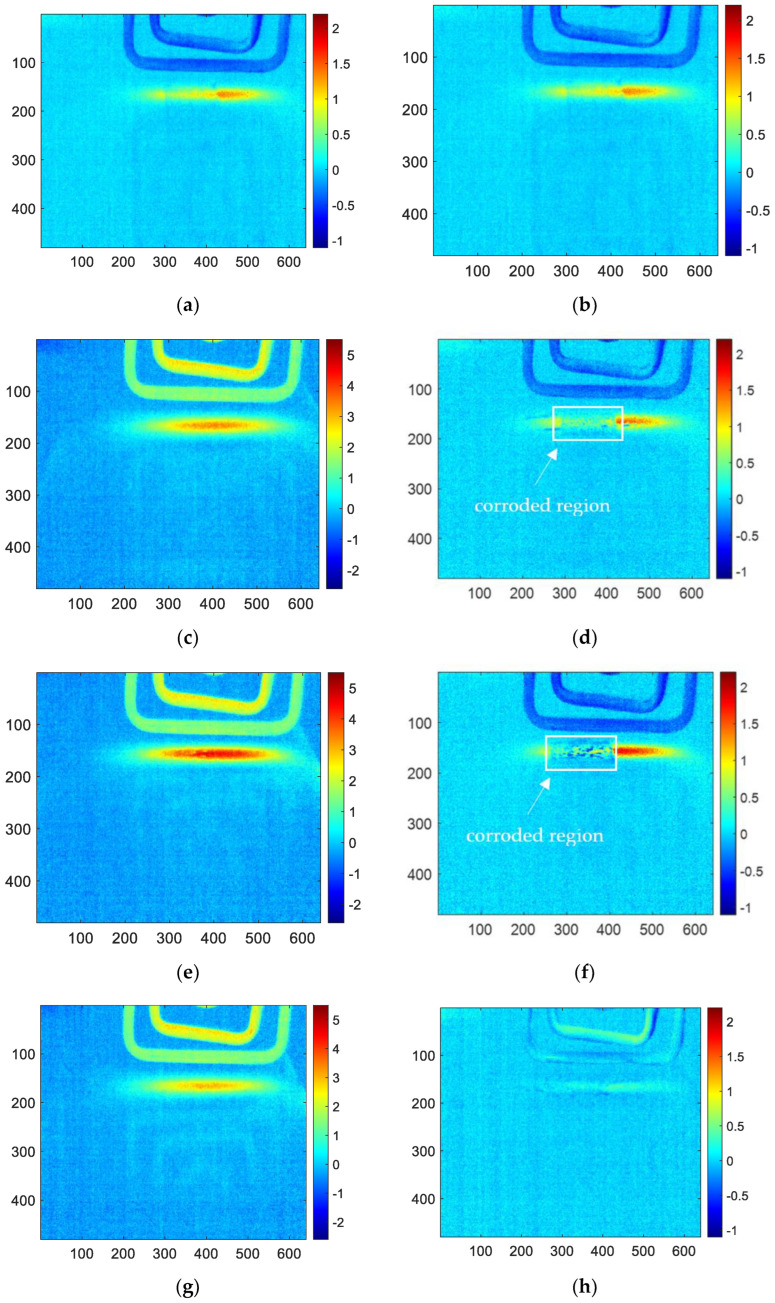
The reconstructed images of (**a**) first PC for 1-month corrosion, (**b**) second PC for 1-month corrosion, (**c**) first PC for 3-month corrosion, (**d**) second PC for 3-month corrosion, (**e**) first PC for 6-month corrosion, (**f**) second PC for 6-month corrosion, (**g**) first PC for 10-month corrosion, (**h**) second PC for 10-month corrosion, (**i**) first PC for 12-month corrosion, (**j**) second PC for 12-month corrosion.

**Figure 10 sensors-23-06889-f010:**
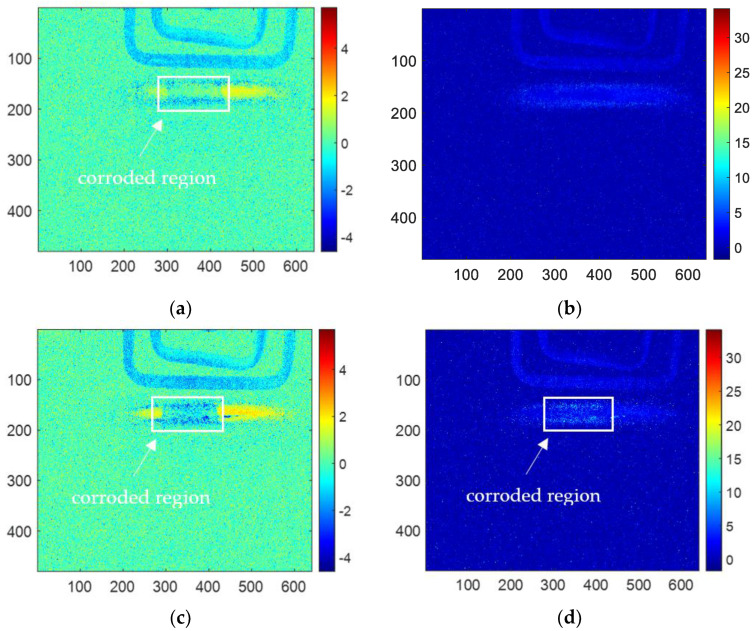
The reconstructed image of (**a**) skewness for 1-month corrosion, (**b**) kurtosis for 1-month corrosion, (**c**) skewness for 3-month corrosion, (**d**) kurtosis for 3-month corrosion, (**e**) skewness for 6-month corrosion, (**f**) kurtosis for 6-month corrosion, (**g**) skewness for 10-month corrosion, (**h**) kurtosis for 10-month corrosion, (**i**) skewness for 12-month corrosion, (**j**) kurtosis for 12-month corrosion.

**Figure 11 sensors-23-06889-f011:**
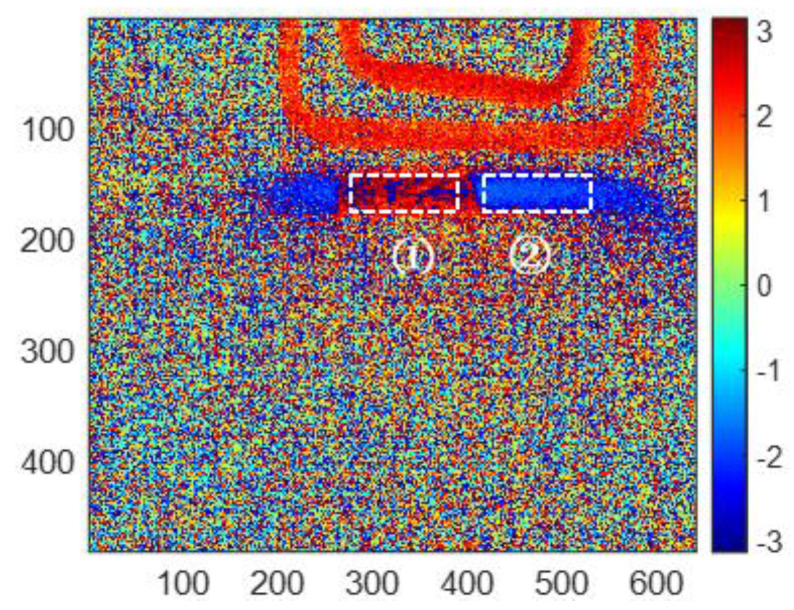
Phase frequency image of 4 Hz with FFT feature extraction. ① is the corroded region, ② is the uncorroded region.

**Figure 12 sensors-23-06889-f012:**
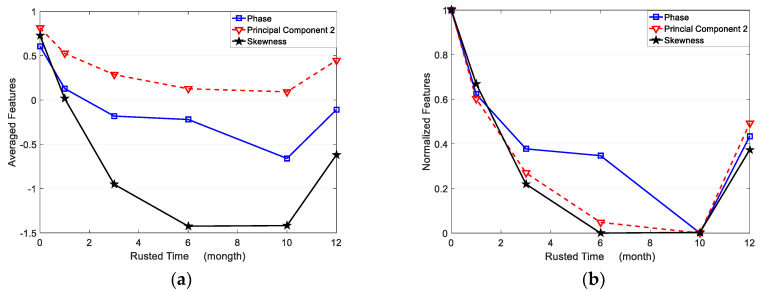
(**a**) Averaged features versus exposure time, (**b**) Normalized features versus exposure time.

**Figure 13 sensors-23-06889-f013:**
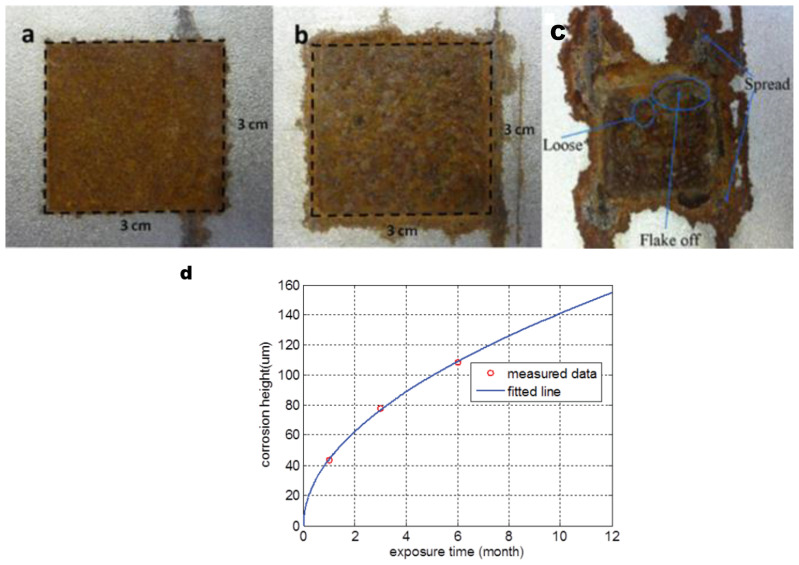
Image of corrosion state: (**a**) 1 month sample, (**b**) 6 months sample, (**c**) 10 months sample, and (**d**) heights versus exposure time [14].

**Table 1 sensors-23-06889-t001:** The composition of S275 steel (wt%).

C	Si	Mn	Ni	S	P	Cr	N	Cu
<0.22	0.05–0.15	<0.65	<0.3	<0.05	<0.04	<0.3	0.012	<0.3

**Table 2 sensors-23-06889-t002:** The averaged features of the corroded region.

The Mean Value of Feature of Corrosion Region	0 Months	1 Month	3 Months	6 Months	10 Months	12 Months
Phase (4 hz)	0.6060	0.1286	−0.1816	−0.2201	−0.6595	−0.1101
second PC	0.8128	0.5262	0.2867	0.1269	0.0922	0.4473
skewness	0.7283	0.0156	−0.9512	−1.4242	−1.4176	−0.6191

**Table 3 sensors-23-06889-t003:** The maximum temperature variation of the uncorroded and corroded region.

Temperature Variation	1 Month	3 Months	6 Months	10 Months	12 Months
Corroded region	0.4920	0.4699	0.6850	0.4183	0.7162
Uncorroded region	0.4678	0.4383	0.7037	0.4126	0.7194

## Data Availability

The data used to support the findings of this study are included within the article.

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
