# Peer review of "Comparison Research on Characterization and Evaluation Approaches for Paint Coated Corrosion Using Eddy Current Pulsed Thermography"

_sensors, 2023, doi:10.3390/s23156889_

Round 1

Reviewer 1 Report

-Thank you very much for the interesting paper. I have some minor comments and questions as follows:

-Please describe the experimental setup in more detail and how the excitation frequency is controlled/modulated. Harmonic, a short excitation burst pulse or a longer heating phase (long burst). This is not clear.

-The explanations to "Fast Fourier transform" 2.1 can be possibly extended with the following source [1], since in my opinion the analysis of pulsed excited thermographic data takes place here.

[1] X. Maldague and S. Marinetti, ‘Pulse phase infrared thermography’, Journal of Applied Physics, vol. 79, pp. 2694–2698, Mar. 1996.

-The reference to Fig11 in text seems to be ambiguous. (region in Fig.10)

„Considering the infrared thermal imaging data are interfered by the instrument itself, environmental temperature and other factors, the features in the corroded region (region in Fig.11) and uncorroded region (region in Fig.10)“

Furthermore, the marked regions 1 and 2 may not be correctly assigned/shifted to the right. Please check or show clearly in relation to fig 8-10.

Good luck with finishing the paper. I am looking forward to read the final paper.

Author Response

Many thanks for your kind help, which enhances the quality of this manuscript obviously. Please check the detailed response in the attached document.

Reviewer 2 Report

Overall some of the sentences are hard to understand, I think that the paper should be proofread, preferably by the native speaker.

Author Response

(The authors gave the same response as above.)

Round 2

Reviewer 2 Report

I would like to thank the authors for all the answers. I think, that the article has been largely improved and can be accepted in the current form.

I would advice the proofreading of the whole article by the native speaker.